# Putative Role of Arthropod Vectors in African Swine Fever Virus Transmission in Relation to Their Bio-Ecological Properties

**DOI:** 10.3390/v12070778

**Published:** 2020-07-20

**Authors:** Sarah I. Bonnet, Emilie Bouhsira, Nick De Regge, Johanna Fite, Florence Etoré, Mutien-Marie Garigliany, Ferran Jori, Laetitia Lempereur, Marie-Frédérique Le Potier, Elsa Quillery, Claude Saegerman, Timothée Vergne, Laurence Vial

**Affiliations:** 1UMR BIPAR, Animal Health Laboratory, INRAE, ANSES, Ecole Nationale Vétérinaire d’Alfort, Université Paris-Est, CEDEX, 94701 Maisons-Alfort, France; 2UMR ENVT-INRA INTHERES, National Veterinary School of Toulouse, 31300 Toulouse, France; emilie.bouhsira@envt.fr; 3Sciensano, Scientific Direction Infectious Diseases in Animals, 1050 Bruxelles, Belgium; nick.deregge@sciensano.be; 4French Agency for Food, Environmental and Occupational Health & Safety, 14 rue Pierre et Marie Curie, CEDEX, 94701 Maisons-Alfort, France; johanna.fite@anses.fr (J.F.); florence.etore@anses.fr (F.E.); elsa.quillery@anses.fr (E.Q.); 5Fundamental and Applied Research for Animal and Health (FARAH) Center, University of Liège, B-4000 Liège, Belgium; mmgarigliany@uliege.be (M.-M.G.); laetitia.lempereur@health.fgov.be (L.L.); claude.saegerman@uliege.be (C.S.); 6UMR Animal, Santé, Territoires, Risque et Ecosystèmes (ASTRE), CIRAD-INRAE-Université de Montpellier, 34398 Montpellier, France; ferran.jori@cirad.fr (F.J.); laurence.vial@cirad.fr (L.V.); 7Unité de Virologie Immunologie Porcines, Laboratoire de Ploufragan/Plouzané/Niort, Anses, 22440 Ploufragan, France; marie-frederique.lepotier@anses.fr; 8UMR ENVT-INRA IHAP, National Veterinary School of Toulouse, 31300 Toulouse, France; timothee.vergne@envt.fr

**Keywords:** African swine fever virus, arthropods, transmission, ticks, *Culicoides*, mosquitoes, biting flies, sand flies, lice, fleas

## Abstract

African swine fever (ASF) is one of the most important diseases in Suidae due to its significant health and socioeconomic consequences and represents a major threat to the European pig industry, especially in the absence of any available treatment or vaccine. In fact, with its high mortality rate and the subsequent trade restrictions imposed on affected countries, ASF can dramatically disrupt the pig industry in afflicted countries. In September 2018, ASF was unexpectedly identified in wild boars from southern Belgium in the province of Luxembourg, not far from the Franco-Belgian border. The French authorities rapidly commissioned an expert opinion on the risk of ASF introduction and dissemination into metropolitan France. In Europe, the main transmission routes of the virus comprise direct contact between infected and susceptible animals and indirect transmission through contaminated material or feed. However, the seasonality of the disease in some pig farms in Baltic countries, including outbreaks in farms with high biosecurity levels, have led to questions on the possible involvement of arthropods in the transmission of the virus. This review explores the current body of knowledge on the most common arthropod families present in metropolitan France. We examine their potential role in spreading ASF—by active biological or mechanical transmission or by passive transport or ingestion—in relation to their bio-ecological properties. It also highlights the existence of significant gaps in our knowledge on vector ecology in domestic and wild boar environments and in vector competence for ASFV transmission. Filling these gaps is essential to further understanding ASF transmission in order to thus implement appropriate management measures.

## 1. Introduction

Vector-borne diseases (VBD) in domestic animals and wildlife disrupt ecosystems, impose a significant burden on animal health, and are an impediment to socioeconomic development [1]. To date, socioeconomic and environmental changes resulted in the redistribution of some vector species and/or extended seasonal transmission periods. The (re)emergence of VBD-related epidemics has been reported in new geographic areas, highlighting the urgent need to develop effective methods of surveillance and control [2]. Within this context, a better understanding of VBD epidemiology and the arthropods potentially implicated in their transmission cycles is essential.

African swine fever virus (ASFV) is a DNA virus and the only member of the family Asfarviridae. This virus only infects domestic and wild Suidae and causes variable symptoms depending on its virulence. Infections range from acute and fatal to chronic or even asymptomatic infection [3,4]. The current absence of any treatment or vaccine makes African swine fever (ASF) one of the most important swine diseases due to its severe impact on animal health and significant socioeconomic consequences [5]. ASF was first identified in Kenya in the 1920s [6] and is currently considered endemic in a large part of sub-Saharan Africa. During the second half of the last century, genotype I ASF strains spread into Europe, causing the first epidemic in Portugal, which then spread to other southern and central European countries and later to South America and the Caribbean [7]. Thanks to drastic control measures, ASF was eradicated from Europe in the 1990s, except on the Italian island of Sardinia. In 2007, a different strain of the virus (Genotype II) was introduced into Georgia from Africa, which then spread through the Caucasus into Russia, Ukraine, and Belarus [8]. In 2014, ASF reached the Eastern European Union. In 2018, ASF was reported in Asia, indicating a global spread of the virus [9]. In that same year—900 km from the nearest known source of infection in the Czech Republic—ASF-infected wild boar carcasses were discovered in southern Belgium near the French border, thus highlighting the likelihood that the disease could be introduced into France [10,11].

Different ASFV transmission cycles occur depending on the geographical area [7]. Firstly, in some parts of sub-Saharan Africa, sylvatic transmission occurs between *Ornithodoros moubata* ticks and warthogs (*Phacochoerus africanus*) [12]. The sylvatic cycle between these two natural virus reservoirs is believed to act as a permanent viral source, infecting the domestic pig value chain [13]. Then during the 1957 ASF outbreak on the Iberian Peninsula in Europe, *Ornithodoros erraticus* soft ticks were described to be involved in an intermediate transmission cycle between ticks and domestic pigs (*Sus scrofa domesticus*). In this case, ticks act as a viral reservoir, enabling the virus to persist locally in the environment [14]. Thirdly, the virus can also be transmitted directly between infected pigs and healthy pigs or indirectly via food products of porcine origin in which the virus has persisted, through what is called the domestic transmission cycle [15]. In this latter cycle, which concerns the majority of ASFV infections worldwide, none of the natural disease reservoirs, i.e., *Ornithodoros* ticks and wild African Suidae, are involved. Finally, and more recently, a fourth epidemiological cycle was described as driving ASF transmission and persistence in Europe. This cycle involves wild boars (*Sus scrofa)*, their carcasses, and their habitat, where both direct transmission and indirect transmission occur between wild boars and via environmental contamination from infected carcasses, respectively [9].

While it is well established that the most frequent transmission in Europe occurs via direct and indirect contact between wild and domestic pigs, some observations question the involvement of local arthropod vectors in ASFV transmission. Firstly, the virus continues to spread among wild boar populations in several areas of Northern Europe despite the implementation of control measures. Secondly, several outbreaks of ASF have been reported in pig farms with high biosecurity, where the source of infection remains unknown [16]. Lastly, epidemics in pig farms are seasonal in nature and appear to correspond with the activity periods of several arthropod vectors [17,18]. Therefore, biological virus transmission seems able to occur via arthropod vectors, which means that the hematophagous arthropod could acquire the pathogen during its blood meal, allow its development and/or multiplication, and then re-transmit it during a new blood meal. Mechanical transmission of the virus may also occur, where transmission takes place between two blood meals through contamination of the mouthparts or regurgitation, without any transformation or multiplication of the pathogen within the vector. Finally, passive viral transmission may also occur through contamination of the arthropod body and/or their ingestion by Suidae, which may also concern non-vector arthropods.

The objective of this review was therefore to gather knowledge on potential ASF transmission mechanisms utilized by a number of arthropod vectors present in metropolitan France, namely soft and hard ticks, *Culicoides*, mosquitoes, biting flies, sand flies, lice, and fleas, in link with their bio-ecology. Both active (biological and mechanical) and passive transmission routes were studied, including transmission via ingestion of infected or contaminated vectors. In addition, whether non-hematophagous arthropods are implicated in ASF transmission through ingestion or passive transport was also addressed. 

## 2. Description of the Ecology and Putative Vector Role of Different Genera/Families of Arthropods

### 2.1. Argasidae

Soft ticks from the genus *Ornithodoros* are the only known and demonstrated biological vectors of ASFV [19]. They are nidicolous with a large host spectrum and are adapted to rather dry environments, requiring warmer-than-temperate temperatures for development. Since soft ticks take repeated blood meals during their nymphal and adult life stages, both stages can be implicated in both biological and mechanical pathogen transmission.

In Africa, ASF transmission by *Ornithodoros* soft ticks can be significant in some geographic areas and occurs via both the sylvatic and intermediate cycles [20]. In East and Southern Africa, where the main vector is *O. moubata*, soft ticks predominantly become infected with ASFV during a blood meal taken from viremic warthogs. In European soft ticks, the virus was first identified in the 1960s in *O. erraticus* from Spain [19]. This species is strongly suspected to have been the origin of the last Portuguese outbreak in 1999 five years after ASF was eradicated from the Iberian Peninsula [14]. In each case, due to trans-stadial, transovarial, and sexual transmission of the virus within tick populations, the vectors also served as reservoir for ASFV [21]. *Ornithodoros* soft ticks are rather opportunistic, as DNA of pigs, humans, cattle, sheep, rodents, and poultry has been detected in *O. erraticus* [22]. *O. erraticus* exclusively colonizes the crevices of old houses made of rubble, hence its presence in traditional pigsties [23]. A study by Diaz and co-workers using artificial feeding systems demonstrated that Portuguese *O. erraticus* was competent for the multiplication and maintenance of the European Georgia 2007/1 strain for at least three months [24]. However, in more recent studies, ASF transmission by both *O. erraticus* and *Ornithodoros verrucosus* (a soft tick species present elsewhere in Europe) to pigs could not be demonstrated while it was shown to be successful with *O. moubata* ticks as a positive control. Both tick species were however capable of maintaining viral infections for up to eight months after their infection [25].

Based on the recent classification of Mans and co-workers, species from the *Ornithodoros* genus have never been identified in metropolitan France [26]. Recent modeling studies have also highlighted the absence of favorable zones in France suitable for the installation of *Ornithodoros* sp. ticks, with the exception of the extreme south of the Mediterranean coast [27]. Current climate change may suggest a northward expansion of Mediterranean and subtropical tick species, but studies indicate a reduction in geographical range of some of the *Ornithodoros* genus species, whichever world location was examined (*Ornithodoros porcinus* in Madagascar*, O. verrucosus* in Ukraine, *O. erraticus* in the Iberian Peninsula, and *Ornithodoros sonrai* in West Africa) [23,28,29,30,31,32]. Among the possible explanations is that the very sedentary and inbred character of these ticks may render them poorly adapted to environmental change and expansion. Secondly, many human actions have modified their preferred habitats, such as improvements in housing or the use of acaricides. Finally, the general drying up of the climate can also negatively impact *Ornithodoros* sp. populations despite their nidicolous way of life [31]. Thus, although uncertainty remains, the risk of introduction and expansion of *Ornithodoros* sp. soft ticks able to transmit the ASFV in France can be considered almost zero today.

### 2.2. Ceratopogonidae

Ceratopogonidae are small insects of the order Diptera present on all continents. Among them, nearly 70 species belonging to the genus *Culicoides* are present in metropolitan France, among which approximately 96% are hematophagous. *Culicoides obsoletus* and *Culicoides scoticus* are the most abundant species found in animal farms and are known vectors for bluetongue virus, *Schmallenberg virus*, and African horse sickness virus. *Culicoides imicola*, known to be the most abundant vector species in Africa, is also present in the south of France and is currently expanding toward the north. *Culicoides* are mostly active from mid-April to early November [33,34]. They are exophilic but can sometimes be abundant indoors when animals are housed in the barn [35]. *Culicoides* have a low active dispersion capacity and movements per active flight are estimated not to exceed 1 km [36]. Their major dispersal method however occurs via passive wind transport. They can be carried for distances ranging from a few kilometers to 700 km [37]. For the majority of *Culicoides* species, only adult females are hematophagous and take an approximately 0.1 µL blood meal every three to four days [36,38]. *Culicoides* have been found in pig farms and blood meal analysis demonstrates that they feed on pigs, although they seem to prefer to feed on ruminants and horses [39]. *Culicoides* sustain biological transmission of various pathogens via their blood feeding behavior and more than 50 viruses belonging to Bunyaviridae, Reoviridae, and Rhabdoviridae families have been isolated from different *Culicoides* species. Some of these viruses are of major concern to animal health worldwide, such as African horse sickness virus (AHSV), bluetongue virus, and *Schmallenberg virus* [40]. *Culicoides* can also mechanically transmit viral infectious agents, as experimentally demonstrated for Rift Valley fever virus [41].

The ecology of *Culicoides*, their potential presence in pig farms, and their ability to biologically and mechanically transmit pathogens makes them convincing potential candidates for ASFV transmission. Although they ingest very small blood volumes and, thus, potentially low viral doses from infected animals, both mechanical and biological transmission and infection of healthy animals could be favored by their abundant presence, due to multiple bites and the injection of sufficient viral doses. Finally, in the case of infection, their high passive dispersal capacity could promote virus spread. However, to date, no studies investigated whether *Culicoides* can transmit ASF. 

### 2.3. Culicidae

Culicidae, or mosquitoes, represent a very large family of the order Diptera and are the most important vectors of diseases with public health significance. Sixty-five species of mosquitoes have been described in metropolitan France, mainly belonging to the genera *Aedes* sp. *Culex* sp., and *Anopheles* sp. *Aedes* spp. are diurnal and exophilic, and two species are predominately found in France: *Aedes caspius* and *Aedes albopictus. Ae. caspius* has significant active and passive dispersal, while *Ae. albopictus* has a more limited active dispersion capacity. Passive dispersal through human transportation has however allowed this species to colonize very large areas of the territory [42]. *Culex* spp. prefer to bite at night and have little dispersal beyond breeding sites (up to 2–3 km). *Culex pipiens* is the predominant species in France. *Anopheles* spp. are present throughout France and are mainly found in marsh and paddy regions. Their dispersal ranges over a few kilometers, and they essentially bite at dawn and dusk. Only females are hematophagous, and the volume of blood ingested varies between 4 and 10 µL from one mosquito species to another [43]. Mosquitoes usually bite every 2 to 3 days before laying eggs; but, if they are interrupted, they can bite several times over a period of a few hours to complete their blood meal. This is why female mosquitoes are able to transmit pathogens from an infected to a healthy host through both biological and mechanical transmission, and this during a lifetime that can extend up to three months under favorable conditions [43,44]. Mosquitoes can mechanically transmit some viruses (myxoma, lumpy skin disease) and bacteria (*Francisella tularensis*) during interrupted meals and contamination of their mouthparts, but they are mostly known as important biological vectors of multiple viruses, protozoa, and nematodes [45]. Blood meal analysis has shown that *C. pipiens, Aedes vexans, Anopheles maculipennis* s.l., *Anopheles claviger*, and *Coquillettidia richiardii* can feed on domestic pigs or wild boars [46,47,48].

Very little information is available on the potential role of mosquitoes in ASFV transmission and no studies have validated any possible biological or mechanical transmission. Only Plowright and co-workers mentioned a study that failed to validate the vectoral competence of mosquitoes and horse flies for the virus [49]. A recent study performed in Estonia detected low viral doses in a mosquito collected from an infected farm, but virus isolation was unsuccessful [50]. These results cannot yet validate a role for mosquitoes in the transmission of ASFV, only the fact that mosquitoes can bite viremic pigs and potentially acquire the virus. Thus, although mosquitoes are recognized as vectors of several viruses and are one of the most abundant vectors in France, there are currently no data available to assess whether they could play a role in ASF transmission. 

### 2.4. Ixodidae

Ixodidae or hard ticks are among the most important vectors of veterinary significance worldwide [51]. In France, about forty different species of ticks are identified, and the main species of importance for both public and veterinary health belong to the genera *Ixodes, Dermacentor,* and *Rhipicephalus* [52]. Species likely to feed on Suidae, mostly at the adult stage, are *Ixodes ricinus, Dermacentor marginatus, Rhipicephalus bursa, Rhipicephalus pusillus, Rhipicephalus turanicus, Haemaphysalis punctata, Haemaphysalis concinna, Haemaphysalis inermis, Hyalomma marginatum,* and *Hyalomma lusitanicum*, and occasionally *Dermacentor reticulatus* and *Pholeoixodes hexagonus* [52]. *Ixodes ricinus* and *D. reticulatus* are the two species most likely to be involved in ASFV transmission in France due to their trophic preferences and their abundance. *Ixodes ricinus* is the most abundant tick in Europe and the most widely distributed. This species is exophilic during all its stages and lives in wooded ecosystems whether rural, peri-urban or even urban, or in pastures [53]. *Dermacentor reticulatus* ticks prefer more humid biotopes such as meadows and forests and can be found in peri-urban areas. Larvae and nymphs are endophilic [54]. Both species usually show peak activity in spring and autumn. Ticks do not disperse by themselves over long distances but can be passively transported by their wild or domestic hosts while attached for their blood meals, which can last up to 10 days for adult life stages. Blood meals of hard ticks are voluminous—more than 1 mL at the adult stage for the majority of species—and have a relatively long duration compared to other hematophagous arthropods. Unlike soft ticks, hard ticks such as *I. ricinus* and *D. reticulatus* only take one blood meal per life stage (i.e., larvae, nymph, adult) and drop off the host after feeding. They then molt into the next stage or lay eggs and die in the case of the adult female. Exceptionally, partial blood meals are taken from different hosts following the death of the host. In general, the life cycles of these three-host ticks can last up to three years during which the tick only takes three blood meals. This lifestyle leaves little possibility for mechanical pathogen transmission and requires trans-stadial transmission for biological transmission to occur.

The ability of hard ticks to transmit ASFV has been assessed for several species, including *I. ricinus* and *D. reticulatus*, either by experimental infection or by detection of the virus in ticks collected from the field. With the exception of DNA segments of ASV recently detected in *Dermacentor* ticks from sheep and bovines in China [55], no other field-collected ticks were found to carry ASFV and transmission could not be demonstrated from experimentally infected ticks [49,56,57]. A recent review on the risk of ASFV transmission by ticks in the Baltic States also concluded that European hard ticks are not involved in potential biological transmission of the virus [58]. Concerning potential mechanical transmission of the virus, as hard ticks only take one blood meal per stage separated by several months of diapause, they cannot, in principle, act as mechanical vectors for ASFV. Alternatively, host infection by ingesting infected ticks could be potentially possible since the virus can be detected several days after experimental infection in some tick species, and oral infection by the virus has been demonstrated [59]. It should also be noted that hard ticks, in case of infection, could transport the virus over long distances while attached to their hosts during their long blood meals.

### 2.5. Muscidae

Within the Muscidae family of Diptera, the genus *Stomoxys* encompasses 18 species, among which *Stomoxys calcitrans* is the only cosmopolitan species present in metropolitan France. In the field, *S. calcitrans* has diurnal and seasonal activity that is dependent on outdoor temperatures. As soon as the outside temperature exceeds 10–11 °C, stable flies become active, with a peak of activity during spring and a longer peak in autumn [60]. Adults live for between two and four weeks on average [61]. They can spend the winter indoors in heated breeding sites and, thus, maintain year-round reproductive activity, nevertheless densities are much higher between May and November [62]. The presence of *S. calcitrans* is primarily associated with livestock farming because it requires decomposing organic plant material and animal manure in which to lay its eggs. Therefore, intensive pig farming systems with a high degree of containment appear less appropriate for *S. calcitrans* development than outdoor farms containing straw [63]. Both sexes are hematophagous. In general, they take one or two meals per day, which can last between 2 and 30 min. Since bites are painful and trigger a host response, between 5 and 20 interrupted meals may be necessary to acquire a full meal. The volume of a blood meal varies between 7 and 15 µL, and the duration between meals can range from four hours to four days [64]. If the flies do not find enough hosts upon which to feed, they can travel great distances of up to 5 km, to find new hosts and more favorable conditions [65]. *Stomoxys* spp. are biological vectors of *Habronema microstoma*, a tropical nematode infecting horses, and mechanical vectors of many pathogens including helminths, protozoa, bacteria, and viruses [64]. Mechanical transmission occurs via residual blood present on the mouthparts, which is then transferred to new animals as the result of interrupted feeding. Alternatively, *S. calcitrans* flies are also known to retain blood in the crop that can then be regurgitated during the next blood meal [64,66]. *S. calcitrans* has been observed to store blood in its crop for over 24 h before it is partially regurgitated—up to 2 µL—or digested in the midgut [67].

No studies have been undertaken on the possible biological transmission of ASFV through *Stomoxys* sp. In contrast, by the late 1980s, it was reported that *S. calcitrans* could mechanically transmit ASFV to susceptible pigs when infected 1 or 24 h earlier [68]. The virus survived in these flies for at least two days without any apparent loss of virulence [68]. Another recent study demonstrated the presence of viral DNA on the mouthparts of these flies for at least 12 h and on the head and body for up to three days after feeding on ASF-contaminated blood in vitro. Infectious virus was found in the body of the flies at 3 and 12 h post-infection [69]. The persistence of high virus titers in flies for periods of up to two days strongly suggests that transmission is possible for at least that length of time [64]. Based on this information, it is then possible that these flies could be involved in the transmission of ASFV between Suidae in France. A recent study performed in Lithuania during summer, reported the presence of *S. calcitrans* in forest areas near pig farms that were frequented by wild boar [70]. This highlights the potential for ASF introduction into pig farms with outdoor access. In addition to further viral spread via direct contact, mechanical transmission by *S. calcitrans* could accelerate disease spread between pigs, which could shorten the time between the introduction and the occurrence of the epizootic peak. Such a phenomenon could then significantly impact the infectivity of the livestock farm and, therefore, the risk of spreading the virus to other areas.

### 2.6. Phlebotominae

Sand flies are small insects of the order Diptera (2–5 mm) with about 900 species present throughout the world. Two genera of Phlebotominae are present in Europe: *Phlebotomus* and *Sergentomyia*. In metropolitan France, they are mainly found in the southern half of the country, with very high population densities in the Mediterranean region. However, recent studies have shown their spread to the north [71]. Their period of activity is from May to October with 1–3 generations per year, and adult longevity is between two weeks and two months. Their active dispersal does not extend beyond 1 km from their breeding/resting sites [72,73]. Only females are hematophagous and are active at dusk and during the night when the temperature reaches 19–20 °C. The interval between each meal, which lasts between 30 s and 5 min, is about 3 to 5 days. Interrupted meals are not documented for sand flies, but some blood meal analyses raised the hypothesis that a female, if disturbed during feeding, can complete her blood meal on a different host [74,75]. Although the majority of the species are exophilic, some of them may be attracted by low-intensity light and, thus, enter livestock buildings or shelters [76]. Various studies, notably from South America and Asia, report that pigs appear to be among the preferred hosts for certain sand fly species [77,78,79,80]. A study in China reported high numbers of engorged sand flies on pigs and hypothesized that soil enriched with organic matter in pig pens could be an appropriate substrate for their larval development [81]. However, it should be noted that all these studies concern exotic sand fly species that are absent from Europe.

Sand flies are predominantly known for transmitting leishmaniasis to humans and animals, but they are also responsible for the transmission of many viruses [82]. American species have been shown to be involved in the transmission of vesiculoviruses responsible for epizootic vesicular stomatitis affecting cattle, pigs, and horses [83]. There are no data in the literature regarding possible transmission of ASFV by sand flies. Although studies have shown that some exotic species can feed on pigs, their exophilic and thermophilic character suggests that they probably do not play a role in the transmission of ASFV in metropolitan France.

### 2.7. Phthiraptera

The order of Phthiraptera comprises more than 5000 species divided into two sub-orders: sucking lice (Anoplura) and biting or chewing lice (Mallophaga). These permanent and obligatory ectoparasites have strict host specificity. *Haematopinus suis*, belonging to the Anoplura sub-order, is the only louse that infests Suidae [84]. Geographically, this louse is present wherever pigs are present but is more common in colder climates of the Northern Hemisphere including Europe and metropolitan France [85]. Its prevalence and abundance, without marked seasonal variation, depend on the type of animal housing conditions and the degree of animal confinement, since they dislike heat and direct sunlight. Under optimal conditions, females lay three to six eggs per day, and the entire cycle from egg to adult, which lasts from three to four weeks, is completed on the host. Adults only survive for a few hours outside of their hosts. They live for about one month, and there can be 6–12 generations per year [85]. Lice remain fairly stationary on their hosts and show little inclination to leave them spontaneously [86]. All stages (three larval stages and one adult stage) and both sexes of *H. suis* are hematophagous. They take small blood meals of approximately 0.1 µL for 10 to 15 min at close intervals or even every hour [87,88]. *Haematopinus suis* is a known vector of porcine smallpox virus, *Mycoplasma suis*, and classical swine fever virus (CSFV) [89]. Despite the low mobility of lice, transmission of pathogens is possible when lice are passed from one animal to another during close contact, for example during mating, lactation (from mother to offspring), accessing the feeder, or in instances of shared material or high animal density. Thus, the introduction of an infested animal into an animal enclosure can be followed by the infestation of all animals within the same lot in only a few days [85].

Only one study has reported the potential role of lice in ASFV transmission. In this study from the 1960s, ASFV was detected in *H. suis* collected from domestic pigs experimentally infected with ASFV [90]. The authors showed that the virus remained virulent for periods greater than 42 days in lice. They also demonstrated that lice acquiring the infection by feeding on a viremic pig could transmit the virus to a healthy pig, which died 42 days after the infestation. This indicates that lice are capable of transmitting ASFV from one pig to another and supports further studies on this potential vector. ASF transmission by lice might, however, be of little epidemiological importance compared to direct contact transmission between pigs given the sedentary nature of this vector.

### 2.8. Siphonaptera

Fleas are small (0.5–8 mm) wingless cosmopolitan insects that belong to the order of Siphonaptera, which comprises about 2500 species and subspecies. They can infest a wide range of mammalian and bird hosts, but studies investigating the presence of fleas on Suidae are scarce. The flea species most often found on pigs, *Sus scrofa domesticus*, in Europe and in Metropolitan France, belong to the Pulicidae family and are *Ctenocephalides felis* (the “cat flea”) and *Pulex irritans* (the “human flea”) [91]. To the best of the author’s knowledge, no studies report the presence of fleas on wild boars, *Sus scrofa*
*scrofa*. Fleas are very sensitive to desiccation but are resistant to cold. They are homogenously distributed throughout Europe and France, although they prefer mainly warm and humid environments without direct sunlight. Their presence and abundance in a given environment can be seasonal, with numbers increasing significantly during the warm summer months [92]. *Ctenocephalides felis,* which may be widespread in confined ruminant farms, is a very sedentary species, and adult fleas do not tend to leave their host. Flea transfer between individual animals can however occur, especially when animals live closely together or when their host’s body temperature drops, as during anesthesia or death [93]. Although wild canids are the reservoirs of *P. irritans*, this flea species is more and more often found in sheep and goat farms and has also been described in pigs [94]. The biology of this species is less well known, but it seems that the adults are less sedentary than those of *C. felis* and are present on the host only during the blood meal process (E. Bouhsira, unpublished results). In particular, it has been shown that this species can be transported by humans in their clothes [95]. The flea development cycle from egg to pupa takes place in protected environments such as barns, burrows, and houses and in close proximity to the infested host. Infestations take place when different host species share the same direct environment. Infestation of hunting dogs by temporarily free-living questing fleas from the vegetation has been described but is really rare, as adult fleas from many species do not survive well off the host [96]. Dogs and other predators can acquire several uncommon flea species from their prey, acquired during hunting activities [96].

Males and females are hematophagous, and the frequency of blood meals varies, depending on the species, from four per day for those living on the host, to one meal every three to four days for those who live in burrows or nests. They can probe multiple times before choosing the most suitable place for their blood meal during which they ingest a volume of between 1 and 1.5 µL of blood over two to ten minutes [97].

Fleas can transmit many pathogens of medical and veterinary significance, such as parasites, bacteria, and viruses, according to different modalities: by bites (through both biological and mechanical transmission), via feces, by regurgitation, or when the host ingests infected fleas [98]. However, there are no data in the literature regarding the possible transmission of ASFV by fleas. It would therefore be interesting to investigate the competence of fleas to transmit ASFV, but their sedentary nature, the lack of reports of infestation of wild boars with fleas, and their low survival off the hosts make it unlikely that they could be involved in the introduction of ASFV to domestic pigs from wild boars or play a major role in its spread between animals.

### 2.9. Tabanidae

Tabanidae, also known as horse flies, are relatively large (6–30 mm) insects that include more than 4000 species worldwide with extensive morphological and biological diversity. In metropolitan France, 83 species are present and distributed over seven genera: *Silvius, Nemorius, Chrysops, Hybomitra, Atylotus, Tabanus,* and *Haematopota* [99,100]. Tabanids are exophilic and prefer rural environments. Depending on the species, the larvae can grow in mud, riverine vegetation of marshes and ponds, under the rocks of streams, or under the bedding of forests [101]. Adults live for between two and four weeks and are active between the end of May and August–September in Europe [99]. Only females are opportunistically hematophagous and feed mainly on the blood of large mammals present in their biotopes [102]. Nevertheless, some species appear to have a stricter host specificity, and several Australian species seem to be specifically attracted to Suidae [103]. This specificity has not been described in Europe, but very few similar studies have been carried out to date. Tabanids generally feed during the day, but their activity is highly dependent on climatic conditions, and some horse flies species respond differently to weather fluctuations [104]. Tabanids take large blood meals compared to most other arthropod vectors, with blood meal size ranging from 20 to 600 µL depending on the species. Long intervals of between five to seven days occur between two complete blood meals [105]. However, due to their painful bites and the subsequent host response, disturbed flies must perform many interrupted meals, which may promote potential mechanical pathogen transmission. Their ability for mechanical transmission is further strengthened by their large mouthparts that allow them to collect greater volumes of blood and which retain large volumes of residual blood that can then be inoculated during a subsequent feeding attempt.

Females are powerful flyers, and when actively searching for a host, they can cover several kilometers [101]. Barros and co-workers showed that after the interruption of their meal, 40% of the horse flies in Brazil returned to the horse of origin while the percentage of partially engorged individuals passing from one horse to another horse at 5 and 25 m was 11% and 4.6%, respectively. No transfer was observed at a distance of 50 m [106]. During a study evaluating the possible spread of equine infectious anemia virus, Lempereur and co-workers observed maximum flight distances of 100 and 200 m when *Haematopota* spp. were partially or totally engorged, respectively [107]. Tabanids are biological vectors of pathogens such as *Loa loa* (filariosis) and *Trypanosoma theileri* but are mainly known to be mechanical vectors of viruses such as equine infectious anemia virus, bacteria such as *Anaplasma marginale*, or some protozoa [108,109].

There is no published evidence on the role of Tabanidae in the transmission or spread of ASFV although it has been repeatedly suggested by some authors [16,110,111]. Their large host tropism and the existence of many interrupted meals indeed suggest a high potential for the mechanical transmission of the virus between infected and healthy pigs. Their ecological preferences for wild and rural biotopes, however, suggest that they could more likely be involved in transmission between wild boars. Nevertheless, their possible involvement in transmission of the virus from wild mammals to pig farms should be considered, in particular, due to their high dispersal capacity. In this respect, several tabanid species (*Haematopota* spp., *Chrysops* spp., *Hybomitra* spp.) have been collected in Lithuania, from forested areas in the vicinity of pig farms and also frequented by wild boar during the summer period [70]. However, their appetite for domestic pigs will depend on the availability of other hosts nearby. In conclusion, horse flies could potentially play a role in the mechanical transmission of ASFV in France within wild boar sounders and perhaps from infected wild boars to neighboring domestic pig farms, thus, it is necessary to carry out vector competence experiments to verify this hypothesis. To do so, laboratory tabanid colonies would need to be developed, which has never been successfully achieved thus far. Furthermore, it is important to investigate the trophic preference of horse flies for wild and domestic Suidae by blood meal analysis.

## 3. Passive Transport ASFV Transmission by Arthropods

ASFV can be passively transported from an infected to a healthy animal by non-hematophagous arthropods. Carrier arthropods can collect virus particles while feeding or laying on a carcass of an infected animal or by contact with infected live animals via wounds, mucous membranes, or bodily fluids (feces, tear secretions, etc.). Such passive transmission of several different viruses has been established. Porcine reproductive and respiratory syndrome (PRRS) virus transmission was confirmed from a sick to a healthy pig by domestic flies and transport of the virus from one farm to another, over a distance of 120 m [112]. Passive transmission has also been demonstrated for H5N1 avian influenza virus found to be infectious on legs of the domestic fly *Musca domestica* and in its droppings [113]. However, to date, there have been few studies investigating the potential passive transport of ASF virus by arthropods.

ASF is known to persist for a long time in the carcasses of Suidae during the decomposition process [114,115]. Under natural conditions, these carcasses are colonized by a wide range of necrophagous arthropods mainly belonging to the orders Diptera and Coleoptera. The species vary over time, and they feed on and lay their eggs into the organic matter [116]. Several of these necrophagous fly species can detect a corpse 10–15 km away and can travel several kilometers a day in search of a laying site. If they carry the remains of infected organic matter on their legs and visit several locations to search for a suitable oviposition site, they could contribute to ASFV spread. Domestic flies (*Musca domestica*), which are not originally necrophagous insects, can also occasionally feed on decomposing organic matter. They are able to cover distances from several kilometers up to 32 km [117], potentially transporting viruses present in organic matter over long distances.

Regarding the possible contamination of arthropods from alive, infected animals, ASFV is excreted through many routes, including nasal, genital, and ocular mucosa [118,119]. In a field study, Pitkin and co-workers observed prolonged contact between farmed pigs and domestic flies feeding on feces and urine, as well as on skin abrasions, and nasal, tear, and salivary secretions from diseased pigs [112].

As mentioned above, this route of transmission was shown to support PRRSV transmission, thus it seems plausible that it could also occur for the far more resistant ASF virus. It would thus be interesting to study the quantity and persistence of the ASFV carried on the various body parts of any potential carrier arthropods and subsequently, to determine whether this dose is consistent with the infectious dose required to infect a healthy animal through contact with exposed body mucosae or via ingestion.

In fact, ASFV transmission via an oral route was documented as early as 1967 [120]. Recently, it was shown that pigs ingesting 20 ASFV-infected *S. calcitrans* mixed with their diet developed an infection [110]. Although arthropods do not comprise a major part of the principal diet of wild boars [121], the latter may occasionally ingest species of necrophagous arthropods during instances of cannibalism [122]. On the other hand, it has also been shown that wild boars sniff carcasses, chew bones, and turn over the soil in search of entomofauna, thus all these activities could also represent a risk of ASFV infection for wild boars [123]. Forth and co-workers however, showed that Calliphoridae larvae feeding on an infected carcass quickly inactivated ASFV [124]. Thus, the infection of wild boar or pigs by ingestion of infected arthropods seems possible, but likely corresponds to exceptional events without epidemiological importance.

## 4. Conclusions

It is well established that direct and indirect contact between infected and non-infected Suidae are the main routes of ASFV transmission in Europe. Only the role of *Ornithodoros* ticks in ASF spread in the sylvatic cycle in Africa is well established, while the role of other arthropod vectors remains largely unstudied. The highest probability of transmission involving arthropods is probably linked to the mechanical vector pathway involving biting flies, particularly in the context of introducing new susceptible pig populations following eradication procedures in infected farms or by increasing the transmission rate in an infected pig population. In this second case, the presence of hematophagous arthropods may effectively contribute to the speed of virus transmission, but it is likely that this role is of lesser importance compared to direct transmission between Suidae. This review thus provides an overview of the current knowledge on ASF transmission by arthropod vectors in relation to their bio-ecological properties. In particular, this review highlights total lack of necessary information on the potential vector competence of difference vectors for ASFV and the absence of bio-ecological knowledge on arthropods in proximity to domestic and wild Suidae. Therefore, future studies are urgently needed to fill these gaps to improve control of the current ASFV epidemic in Europe. Indeed, although this study focuses on metropolitan France, it is realistic to extend the conclusions to central and southern Europe.

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
