# Peer review of "Putative Role of Arthropod Vectors in African Swine Fever Virus Transmission in Relation to Their Bio-Ecological Properties"

_viruses, 2020, doi:10.3390/v12070778_

Round 1

Reviewer 1 Report

This is a very interesting, useful and thorough review of the literature. There are a few references that might also (but not necessarily) have been included:

Chen et al. (2019). DNA segments of African Swine Fever Virus detected for the first time in hard ticks from sheep and bovines.  DOI: 10.11158/saa.24.1.13

Endris and Hess (1994). Attempted transovarial and venereal transmission of African swine fever virus by the Iberian soft tick Ornithodoros (Pavlovskyella) marocanus (Acari: Ixodoidea: Argasidae).   DOI:10.1093/jmedent/31.3.373

Golnar et al. (2019). Reviewing the potential vectors and hosts of African swine fever virus transmission in the United States. doi:10.1089/vbz.2018.2387

I have some minor comments below

Point 9 (fleas): Is there any documentation (other than anecdotal) of flea species infesting wild boar? It would seem that there would be a risk of transmission from flea-infested and ASF infected wild boar in close proximity to domestic pigs. Whether known or not, it may be worth specifically mentioning wild boar in this section.

There are some statements that I would suggest re-phrasing for more scientific stringency:

Line 128 – instead of “studies have shown… were not able to transmit” this could be expressed as “ in more recent studies…. transmission could not be demonstrated…” It would also be useful to mention that this experiment used O moubata as control and successfully achieved transmission from this species.

Line 136 – I suggest changing from “field surveys… report a reduction…” to “studies…. indicate a reduction” as many of the cited references cannot be classified as proper surveys of tick prevalence (and 2 of them use the same data).

Line 232 – instead of stating that “no experimentally-infected ticks were able to transmit”, I suggest something less categorical in the line of “transmission could not be demonstrated from experimentally infected ticks”.

Language comments:

Line 182 – I suggest changing from “territorial areas of the territory” to either “territorial areas” of “areas of the territory”.

Line 229 – please replace “for” by “of” (the ability of hard ticks)

Line 342 – please replace “there” by “they” (They are homogenously distributed…)

Line 448 – please replace “principle” by “principal” or “main” (main diet)

Line 467 – “critical lack of information” sounds a little odd, I suggest rephrasing to “total lack of necessary information”

There are also some proofing errors as regards citations in the text (e.g. lines 127, 134, 153, 154, 195, 397) and references 31 and 72 should be checked for completeness, and capital letter of country/city of origin.

Author Response

This is a very interesting, useful and thorough review of the literature. There are a few references that might also (but not necessarily) have been included:

Chen et al. (2019). DNA segments of African Swine Fever Virus detected for the first time in hard ticks from sheep and bovines.  DOI: 10.11158/saa.24.1.13

Endris and Hess (1994). Attempted transovarial and venereal transmission of African swine fever virus by the Iberian soft tick Ornithodoros (Pavlovskyella) marocanus (Acari: Ixodoidea: Argasidae).   DOI:10.1093/jmedent/31.3.373

Golnar et al. (2019). Reviewing the potential vectors and hosts of African swine fever virus transmission in the United States. doi:10.1089/vbz.2018.2387

Thank you very much for this favorable opinion on our manuscript and for your suggestions. Due to the high number of references already include in the MS (and as the reviewer mentioned that it is not necessarily), we only included the most important one from our point of view, i.e.:

Chen et al. (2019). DNA segments of African Swine Fever Virus detected for the first time in hard ticks from sheep and bovines.  DOI: 10.11158/saa.24.1.13

I have some minor comments below

Point 9 (fleas): Is there any documentation (other than anecdotal) of flea species infesting wild boar? It would seem that there would be a risk of transmission from flea-infested and ASF infected wild boar in close proximity to domestic pigs. Whether known or not, it may be worth specifically mentioning wild boar in this section.

Thanks for your suggestions.

To the best of our knowledge no available publications report the presence of fleas on wildboars, suggesting that they are not considered as ectoparasites of major significance in this host species. This information was added in the new version of the MS.

Adult fleas are considered as sedentary parasites remaining on their hosts (C. felis, C. canis, Echinophaga sp.) or in the direct environment such as burrows and barns of the infested hosts (for Tunga penetrans or for P. irritans). The transmission of adult fleas can occur during close contacts between hosts, or by contact with the infested environment.  Adult fleas are rarely/never found as questing parasites waiting for a host on the vegetation, as the survival off the hosts is very limited. Therefore, their sedentary nature, the lack of reports of infestation of wildboars with fleas, and their low survival off the hosts, make it unlikely that they could be involved in the introduction of ASFV to domestic pigs from wildboars, or play a major role in its spread between animals.These clarifications have been also added to the manuscript.

There are some statements that I would suggest re-phrasing for more scientific stringency:

Line 128 – instead of “studies have shown… were not able to transmit” this could be expressed as “in more recent studies…. transmission could not be demonstrated…” It would also be useful to mention that this experiment used O moubata as control and successfully achieved transmission from this species.

The MS has been corrected as suggested

Line 136 – I suggest changing from “field surveys… report a reduction…” to “studies…. indicate a reduction” as many of the cited references cannot be classified as proper surveys of tick prevalence (and 2 of them use the same data).

The MS has been corrected as suggested

Line 232 – instead of stating that “no experimentally-infected ticks were able to transmit”, I suggest something less categorical in the line of “transmission could not be demonstrated from experimentally infected ticks”.

The MS has been corrected as suggested

Language comments:

Line 182 – I suggest changing from “territorial areas of the territory” to either “territorial areas” of “areas of the territory”.

The MS has been corrected as suggested

Line 229 – please replace “for” by “of” (the ability of hard ticks)

The MS has been corrected as suggested

Line 342 – please replace “there” by “they” (They are homogenously distributed…)

The MS has been corrected as suggested

Line 448 – please replace “principle” by “principal” or “main” (main diet)

The MS has been corrected as suggested

Line 467 – “critical lack of information” sounds a little odd, I suggest rephrasing to “total lack of necessary information”

The MS has been corrected as suggested

There are also some proofing errors as regards citations in the text (e.g. lines 127, 134, 153, 154, 195, 397) and references 31 and 72 should be checked for completeness, and capital letter of country/city of origin.

Thanks for that, changes have been made and we hope that no mistake remains.

Reviewer 2 Report

This paper is well written and very informative.

It would be better with an improved structure of deployment.

I suggest section division : 1. Introduction; 2. Describe ecology and role of arthropod by species; 3. Passive transport ASFV transmission by arthropods; 4. Conclusion.

Contents must be more concentrated on the ASF not on the previous studies (examples) on other diseases.

In addition, presence of a table summarizing contents would help understanding the key message of this paper.

Author Response

This paper is well written and very informative.

Thank you very much for this favorable opinion on our manuscript.

It would be better with an improved structure of deployment.

I suggest section division : 1. Introduction; 2. Describe ecology and role of arthropod by species; 3. Passive transport ASFV transmission by arthropods; 4. Conclusion.

Thank you very much for this very good suggestion. The structure of deployment has been modified as required:

  1. Introduction;
  2. Description of the ecology and putative vector role of different genera/families of arthropods
  3. Passive transport ASFV transmission by arthropods;
  4. Conclusion.

Contents must be more concentrated on the ASF not on the previous studies (examples) on other diseases.

We agree that the focus of this review should be on ASF, and we think this is the case.

However, since very little data on the vector capacity for ASF of several genera/families of arthropods dealt with in this review is available, we find it important to shortly mention what is known on their capacity to transmit other diseases. This supports our view that their vector capacity for ASF cannot be excluded as such and deserves to be studied in more detail.

In addition, presence of a table summarizing contents would help understanding the key message of this paper.

We understand the reviewer’s point of view. However, and as indicated in the manuscript, a lot of data is currently missing in order to make conclusions on the putative role of the different arthropods envisaged as vectors of ASF. In addition, it is also very difficult to make such a generalized table, seen the individual differences between species of specific families and genera for certain biological parameters of the blood-feeding arthropods under consideration. For all these reasons, it seems difficult to present a table containing the conclusions of the manuscript that does not seem opportune to us.